# The MicroRNA Landscape of Acute Beta Cell Destruction in Type 1 Diabetic Recipients of Intraportal Islet Grafts

**DOI:** 10.3390/cells10071693

**Published:** 2021-07-04

**Authors:** Geert A. Martens, Geert Stangé, Lorenzo Piemonti, Jasper Anckaert, Zhidong Ling, Daniel G. Pipeleers, Frans K. Gorus, Pieter Mestdagh, Dieter De Smet, Jo Vandesompele, Bart Keymeulen, Sarah Roels

**Affiliations:** 1Diabetes Research Center, Brussels Free University (VUB), Laarbeeklaan 103, 1090 Brussels, Belgium; geert.stange@vub.ac.be (G.S.); zhidong.ling@vub.ac.be (Z.L.); daniel.pipeleers@vub.ac.be (D.G.P.); frans.gorus@vub.ac.be (F.K.G.); bart.keymeulen@uzbrussel.be (B.K.); sarah.roels@colruytgroup.com (S.R.); 2Department of Laboratory Medicine, Molecular Diagnostics Unit, AZ Delta General Hospital, 8800 Roeselare, Belgium; dieter.desmet@azdelta.be; 3Department of Biomolecular Medicine, Ghent University, 9000 Gent, Belgium; Jasper.Anckaert@UGent.be (J.A.); Pieter.Mestdagh@UGent.be (P.M.); jo.vandesompele@ugent.be (J.V.); 4Diabetes Research Institute, Università Vita-Salute San Raffaele, 20132 Milan, Italy; piemonti.lorenzo@hsr.it; 5Center for Medical Genetics, University Hospital Ghent (UZ Gent), De Pintelaan 185, 9000 Ghent, Belgium

**Keywords:** beta cell, type 1 diabetes, islet transplantation, biomarkers, microRNA

## Abstract

Ongoing beta cell death in type 1 diabetes (T1D) can be detected using biomarkers selectively discharged by dying beta cells into plasma. microRNA-375 (miR-375) ranks among the top biomarkers based on studies in animal models and human islet transplantation. Our objective was to identify additional microRNAs that are co-released with miR-375 proportionate to the amount of beta cell destruction. RT-PCR profiling of 733 microRNAs in a discovery cohort of T1D patients 1 h before/after islet transplantation indicated increased plasma levels of 22 microRNAs. Sub-selection for beta cell selectivity resulted in 15 microRNAs that were subjected to double-blinded multicenter analysis. This led to the identification of eight microRNAs that were consistently increased during early graft destruction: besides miR-375, these included miR-132/204/410/200a/429/125b, microRNAs with known function and enrichment in beta cells. Their potential clinical translation was investigated in a third independent cohort of 46 transplant patients by correlating post-transplant microRNA levels to C-peptide levels 2 months later. Only miR-375 and miR-132 had prognostic potential for graft outcome, and none of the newly identified microRNAs outperformed miR-375 in multiple regression. In conclusion, this study reveals multiple beta cell-enriched microRNAs that are co-released with miR-375 and can be used as complementary biomarkers of beta cell death.

## 1. Introduction

The triggers and kinetics of pancreatic beta cell destruction in type 1 diabetes (T1D) are largely enigmatic. It is thought that insidious episodes of beta cell death induced by infectious, nutritional or environmental stressors trigger progressive bouts of T-cell-mediated destruction in individuals with unfavorable HLA backgrounds, with a more aggressive course in younger subjects. Real-time monitoring of cell death in rare cell types such as the beta cells requires highly selective biomarkers with a sufficiently long half-life in plasma, measured by extremely sensitive analytical techniques. To date, only three chemically distinct biomarkers were independently confirmed in models of synchronous necrotic beta cell destruction such as streptozotocin-induced diabetes in rodents or islet transplantation in human T1D: GAD65 protein, unmethylated insulin DNA and microRNA-375 (miR-375). GAD65, the prototypic T1D autoantigen, can be used to predict the outcome of islet transplantation [1], but its use is limited due to interference by GAD65-autoantibodies and technical limitations of analytical sensitivity of sandwich immunoassay technology [2]. Analytically, nucleic acid amplification assays potentially offer superior sensitivity. Beta cell-selective patterns of DNA methylation provide an acceptable level of specificity, allowing the use of unmethylated insulin DNA to detect acute beta cell destruction [3,4]. However, liquid biopsies for DNA methylation patterns are analytically complex, and inter-laboratory agreement of assays is to date disappointing [5]. Furthermore, it is still unknown whether insulin gene DNA methylation patterns are uniformly present in all beta cells or show variations related to functional and developmental heterogeneity or pathological variations in stressed cells. Multiplexing various DNA loci is likely needed to attain the required specificity [6,7]. miR-375 shows an excellent level of islet enrichment [8], is expressed at very high levels by beta cells [9,10] and can be used to detect beta cell death in animal models and islet graft recipients [11]. We previously reported that miR-375 and GAD65 proteins are expressed at near equimolar levels in human beta cells and are co-released at near equimolar levels with roughly comparable half-lives in the circulation. The plasma levels of GAD65 and miR-375 measured 1 h after intraportal islet transplantation are both good indicators of the amount of early graft destruction and can clinically be used to predict the insulin secretory graft function 2 months later [9]. Using optimized assays with analytical sensitivities in the femtomolar range, incidental surges above the limits of detection (LoD) of GAD65 or miR-375 assays were also observed in the follow-up period between day 2 and day 60 after transplantation [9]. A remaining challenge was the confident attribution of such incidental surges above the LoD to actual events of insidious beta cell death rather than analytical noise around the LoD. We reasoned that one way to circumvent this uncertainty was a multiplexing approach: the simultaneous detection above LoD of multiple beta cell-selective intracellular molecules could thus minimize false discovery rate and enable a more specific screening for occult beta cell death in vivo.

The aim of this study was to identify other beta-cell-enriched microRNAs that are co-released with reference marker miR-375 in the well-controlled model of intraportal islet transplantation in T1D subjects. A stepwise approach was followed to prioritize novel candidate biomarkers (graphically summarized in Figure 1). In a discovery cohort, plasma samples before and 1 h after islet transplantation were profiled for the presence and level of 733 human microRNAs. Second, likely beta cell-derived microRNAs were sub-selected by recovery experiments and co-linearity with GAD65 or miR-375. This resulted in a panel of 15 microRNAs that were then subjected to central blinded verification on independent validation cohorts obtained from two different transplant centers. Finally, a core panel of eight microRNAs including miR-375 was then clinically validated in a third independent cohort of 46 islet transplant events for their potential to quantify acute graft destruction and predict long-term graft function (C-peptide secretion). In sum, our study aims to provide a detailed view of the plasma microRNA landscape in the model of intraportal islet transplantation and serve as a resource of novel candidate biomarkers for beta cell death in vivo.

## 2. Material and Methods

### 2.1. Intrahepatic Transplantation of Human Beta Cells in Type 1 Diabetic Recipients

Plasma was sampled 1 h before and 1 h after intrahepatic infusion of human islet preparations in the University Hospital Brussels, Brussels, Belgium (*n* = 16, Brussels cohorts, Diabetes Research Center, B. Keymeulen) and Ospedale San Raffaele, Milan, Italy (*n* = 8, Milan cohort, Diabetes Research Institute, L. Piemonti). The Brussels cohort was divided into a discovery cohort (*n* = 8) and an independent validation cohort 1 (*n* = 8). MicroRNA profiling by Low-Density Arrays was first done on a discovery cohort (*n* = 8) of GAD65 autoantibody-negative (GADA < 23WHO Units/mL) T1D individuals who received at least 2 × 10 [6] beta cells/kg bodyweight with islet graft compositions detailed in Appendix A. Selected microRNAs with biomarker potential were then integrated into a targeted LDA RT-PCR panel, which was verified on an independent validation cohort 1 (*n* = 8) in Brussels and validation cohort 2 in Milan (*n* = 8). For blinded analysis, all pre-and post-transplant patient samples and healthy controls received a random numerical code within each cohort. All samples were then centrally analyzed by PCR and only after finalization of data processing with normalization were samples unblinded. Transplantations in the Brussels cohorts were done using cultured human islets pooled from one to six donors, characterized for beta cell number, endocrine purity and viability by immunohistochemistry (insulin, glucagon), electron microscopy and propidium iodide counting [12]: an average of 9.7 (95% CI: 8.4–11.0) million cells/kg bodyweight, of which 2.9 (95% CI: 2.5–3.3) million/kg bodyweight insulin-positive cells were implanted. Grafts were composed of 40% (95% CI: 34–47) endocrine cells with 32% (95% CI: 27–36) beta cells and 12% (95% CI: 9–16) alpha cells. The rest of the graft consisted of 47% (95% CI: 41–54) non-granulated (mainly ductal) cells, 2% (range 0–8%) acinar exocrine cells and 10% (range 7–14) dead cells (Appendix A). Patients received immune suppression using Anti-Thymocyte Globulin, mycophenolate mofetil and tacrolimus (protocol NCT00623610, ethical approval 98/059D) [12,13]. Islet isolation and intrahepatic transplantation in the Milan cohort were done according to similar protocols albeit with shorter pre-transplant culture times [14]. Plasma samples were collected in K3-EDTA Monovette tubes (Sarstedt, Nümbrecht, Germany) supplemented with 1% of a 0.12 mg/mL solution of aprotinin (Stago, Asnières sur Seine, France) in 0.9% NaCl. Samples were stored at −80 °C after centrifugation for 15 min at 1600× *g*.

### 2.2. Recovery—Linearity Experiments with Human Beta Cell Lysate

Endocrine-enriched islet preparations (52% insulin positive, pool of 3 donors) were washed 3 times with PBS to remove albumin and resuspended in RIPA buffer supplemented with proteinase inhibitor cocktail PIC 1 and PIC 2 (Roche, Basel, Switzerland) at a final cell density of 10^4^ cells/µL. Lysis was done by freezing at −80 °C, thawing on ice, sonication and collection of supernatants after centrifugation for 10 min at 4 °C at 10,000× *g*. This lysate was then spiked to K3-EDTA human plasma to obtain the equivalent of 5, 50 and 250 × 10^3^ cells/mL (K cells/mL).

### 2.3. GAD65 Cytometric Bead Array and Calibrated miR-375 Assay

GAD65 was measured by an in-house developed Cytometric Bead Array (CBA) [2,15] with Limit of Detection (LoD) and Limit of Quantification (LoQ) of 0.121 pmol/L and 0.288 pmol/L respectively. miR-375 was quantified using a calibrated hydrolysis probe-PCR after sequence-specific capture by hybridization using the Taqman miRNA ABC Purification Kit Human Panel A [9]. LoD and LoQ of this miR-375 assay are, respectively, 0.049 pmol/L and 0.102 pmol/L.

### 2.4. microRNA Array Profiling

MicroRNAs were extracted from human plasma using the Taqman miRNA ABC Purification Kit Human Panel A and Panel B (Applied Biosystems, Waltham, MA, USA). microRNAs were measured by hydrolysis-probe PCR with Taqman Megaplex Pools consisting of matching primer pools and Taqman MicroRNA Arrays for 733 human microRNAs according to manufacturer’s protocol (Applied Biosystems, Waltham, MA, USA). microRNA was converted to cDNA using Megaplex RT Primers Human Pool Set v3.0 and pre-amplified using Megaplex PreAmp Primers Human Pool set v3.0. Taqman MicroRNA Arrays (TaqMan Array Human MicroRNA Card Set v3.0) were run on a 7900HT Fast Real-Time PCR System using the 384-well Taqman Array Block with default thermal-cycling conditions (Applied Biosystems, Waltham, MA, USA).

### 2.5. microRNA Array Data Analysis

7900HT Fast Real-Time PCR System output files (.sds) of the discovery cohort samples were analyzed using ExpressionSuite Software v1.0.3 (Applied Biosystems, Waltham, MA, USA). Raw Cq values (quantification cycle, the standard name for Ct according to Real-time PCR Data Markup Language, MIQE and ISO 20395:2019 guidelines [16]) were calculated applying automatic baseline settings, and threshold settings were adjusted for each individual microRNA with equal threshold settings across all samples within this study. Only Cq values equal to or below 35 were considered detectable and taken into account for downstream quantitative analysis. The raw Cq data were further analyzed using qbase+ Software v3.0 [17,18], where raw Cq values were transformed to linear scale (RQ values) and normalization factors were calculated following global mean normalization strategy [19] to obtain one NRQ (normalized relative quantity) value for each microRNA-sample combination. NRQ data were then log10 transformed and uploaded in RStudio (R package version 2.17.0), where an additional data filtering was performed: microRNAs that were detected with low confidence (Cq > 35) in at least 4 of 8, either in the pre-transplant or post-transplant sample group, were excluded. For lysate spiking, a maximum of one missing value was allowed per microRNA. For calculation of fold increase or decrease at group level, missing values of individual microRNAs (Cq > 35) were replaced by the minimum log10(NRQ) minus 1. These final data were converted into a heat map using R package gplots [20] with default clustering settings (complete linkage method for Ward clustering and Manhattan distance measure).

For the profiling of in beta cell recovery experiments, microRNAs were included when the slope of Cq/log (K beta cells/mL plasma) < −1.5; R^2^ > 0.9 and when at least 4/6 criteria (Cq(0 K/mL) > Cq(5 K/mL); Cq(5 K/mL) > Cq(50 K/mL); Cq(50 K/mL) > Cq(250 K/mL); NRQ(0 K/mL) < NRQ(5 K/mL); NRQ(5 K/mL) < NRQ(50 K/mL); NRQ(50 K/mL) < NRQ(250 K/mL)) were fulfilled. Furthermore, the Cq values should progressively decline, and the ∆Cq between subsequent spiked samples ≥ 1.

### 2.6. Targeted PCR in Validation Cohorts

Selected (*n* = 15) candidate biomarker microRNAs (Figure 1) were integrated as singleplex Taqman™ miRNA expression assays in the TaqMan^®^ low-density array format (LDA, Applied Biosystems, ThermoFisher, Waltham, MA, USA) based on sequences provided in Appendix A. The LDA additionally included candidate reference microRNAs (*n* = 4) for data normalization selected from the microRNA profiling data as follows: standard deviation of all NRQ values (for all samples) per microRNA in the microarray analysis of the discovery cohort were calculated. The microRNAs with the lowest NRQ standard deviation and that were detected in at least 15 of 16 samples (*n* = 8, pre- and post-transplant) were imported in qbase+ Software v3.0 (10,11) to determine the 15 best-ranked normalizers (geNorm *M* value, in Appendix A). Selection of the optimal number of stable normalizers (geNorm *V* value) was based on geNorm’s pairwise variation analysis between subsequent normalization factors using a cut-off value of 0.15 for the inclusion of additional normalizers [17]. The genomic location of the four microRNAs (miR-524-3p, miR-30e-3p, miR-208a-3p and miR-1291) candidate reference microRNAs was verified to avoid including microRNAs that are putatively co-regulated [21]. All four candidate reference microRNAs failed in LDA in both subsequent validation cohorts, with inconsistent or no detection. As a consequence, PCR data in the validation cohorts were expressed as RQ instead of reference-gene geometrically normalized values and calculated on paired pre- and post-transplant samples as 2 ΔCt with ΔCt = (mean of duplicate Ct of microRNA*_i_* pre-transplant–mean of duplicate Ct of microRNA*_i_* post-transplant). In the clinical validation cohort of 46 islet transplantations, RQ values were calculated on the post-transplant samples and calculated as 2 ΔCt where Ct = (mean Ct of microRNA*_i_* across all 46 samples—mean of duplicate Ct of microRNA*_i_* in sample X). Tissue selectivity of microRNA expression was analyzed on in-house prepared pancreatic cell fractions obtained by COBE 2991 Cell Processor and enriched in exocrine amylase-expressing acinar cells (1.5% contaminating beta cells), CK19-expressing duct cells (0.4% contaminating beta cells) and endocrine fractions with low (8.3%) or high-percentage (61.0%) insulin-positive cells. Body-wide tissue selectivity was addressed using public repositories of microRNA expression in 61 human tissue types [22] and next-generation sequencing of human islets versus other human tissues [10].

### 2.7. Statistical Analyses

Correlation and linear regression analyses were done using Prism 5 (GraphPad, San Diego, CA, USA). Statistical differences in NRQ or RQ of microRNA levels before and after transplantation were done using two-tailed paired Student’s *t*-test in R without correction for multiple comparison. The area under the Receiver Operating Characteristic (ROC) curve (AUC) (non-parametric model by DeLong et al. [23]), correlogram (Spearman rank) and multiple regression analysis of microRNAs versus graft outcome were analyzed using MedCalc (version 19.2.1, Mariakerke, Belgium).

## 3. Results

### 3.1. Study Design

Intrahepatic infusion of islets is associated with an acute destruction of 2–10% of implanted beta cells, resulting in a synchronous discharge of beta cell-selective biomarkers such as GAD65 and miR-375 [9]. To identify other microRNAs discharged from necrotic graft cells, a step-wise approach was followed as summarized in Figure 1. First, blood samples just before and 1 h after transplantation were profiled by hydrolysis-probe PCR array for the presence of 733 human microRNAs in a discovery cohort of eight patients. After stringent data filtering, with exclusion at sample level (all microRNAs with Cq > 35) and at group level (all microRNAs that were non-detected in at least 4 of 8 samples in either the pre-or post-transplant samples), 220 microRNAs were considered confidently detected and used for further quantitative analysis by global mean normalization and calculation of normalized relative quantities (NRQ) (all data in Appendix A). Of these 220 microRNAs, 29 microRNAs showed significantly different NRQ levels between the pre- and post-transplant phase: seven microRNAs showed lower post-transplant levels, and 22 microRNAs showed a least four times higher post-transplant levels (Figure 1a). Next, these 22 increased microRNA were then further manually down-selected to a panel of 15 candidate microRNAs, guided by following criteria: (i) the magnitude of post-transplant surge (NRQ levels); (ii) a correlation with plasma concentrations of GAD65 and/or miR-375 measured by calibrated assays; and/or (iii) a co-linearity with the amount of spiked-in beta cell lysate in a separate recovery experiment (Figure 1b). The resulting panel of 15 microRNAs (shown in Figure 1b and Table 1) was then subjected to blinded verification by targeted PCR in two independent validation cohorts: eight transplant events in Brussels (validation cohort 1) and seven transplant events in Milan (validation cohort 2) (Figure 1a), resulting in a core panel of eight microRNAs that were subsequently clinically validated.

### 3.2. microRNA Profiling in the Discovery Cohort

Microarray profiling confirmed miR-375 as excellent biomarker of beta cell destruction: in pre-transplant samples, levels were consistently low, and its post-transplant NRQ values (118-times up, *p* = 0.007) correlated well with plasma concentrations of GAD65 quantified by a cytometric bead array [15] (*r_S_* = 0.88; *p* < 0.0001, Figure 1B, Table 1). Besides miR-375, several other microRNAs with a previously reported [8,24] islet endocrine-enriched expression pattern were increased: Let-7g-5p and three members of the miR-376 cluster. The levels of miR-376a-5p and miR-376a-3p correlated positively with GAD65 (*rs* = 0.52; *p* = 0.0396) and with the numbers of implanted beta cells (*r_S_* = 0.81; *p* = 0.0218), respectively. Some of these increased microRNAs might also originate from other cell types contaminating the graft: miR-197-3p correlated positively with the number of duct cells (*r_S_* = 0.9048; *p* = 0.0046), and three microRNAs correlated positively with the number of alpha cells (Figure 2a): miR-125b-5p (*r_S_* = 0.8264; *p* = 0.0154), miR-204-5p (*r_S_* = 0.8623; *p* = 0.0107) and miR-132-3p (*r_S_* = 0.7545; *p* = 0.0368). The miR-375 levels showed a tight hierarchical clustering with miR-200a-3p, miR-369-5p, miR-125b-5p, miR-204-5p, miR-429 and miR-216b-5p (Figure 2a): all showed consistently low baseline levels and marked increases (NRQ range 36- to 6249-fold) 1 h after islet infusion. Of note, intraportal transplantation was not associated with significantly increased levels of hepatocyte-specific microRNA-122 (NRQ = 18, *p* = 0.294) nor muscle-specific miR-133a (NRQ = 7.59, *p* = 0.0619) potentially originating from surgical procedure-associated tissue injury (Appendix A).

### 3.3. microRNAs Correlating Linearly to Beta Cell Number in Recovery Experiment

A restricted set of 16 microRNAs showed a dose-dependent increase proportionate to the number of spiked-in beta cells (Figure 2c). These included miR-375, which showed the strongest correlation with both the number of spiked beta cells and the associated GAD65 concentration (Figure 2d), but also several other microRNAs that clustered tightly with miR-375 in the post-transplant samples, such as miR-125b-5pmiR-429, miR-204-5p and miR-200a-3p (Figure 2a). Overall, 27% (6/22) microRNAs that increased after transplantation showed dose-dependency in the recovery experiment, suggesting their predominant provenance from the grafts (Figure 1b, Figure 2d).

### 3.4. Selection of Candidate Biomarker MicroRNAs for Validation Studies

A panel of 15 microRNAs was sub-selected for blinded validation on two independent validation cohorts. The criteria for selection (Figure 1a) were (i) significant correlation with plasma GAD65 and/or miR-375 levels after transplantation, (ii) increased levels in the recovery experiment proportionate to the spiked-in beta cell number (miR-375, miR-125b-5p, miR-204-5p, miR-429, miR-132-3p and miR-200a-3p), (iii) absence of correlation with duct cell marker miR-197-3p and (iv) one microRNA (miR-103a-3p), because of its consistently low pre-transplant baseline plasma levels and strong correlation to several of the selected microRNAs mentioned above. The 15 candidate biomarker microRNAs selected for validation studies are shown in Table 1, along with their average fold increase (average NRQ ratio in discovery cohort) and correlation with GAD65 and miR-375 concentrations. These 15 microRNAs were integrated into a TaqMan^®^ low-density array (LDA) PCR, along with four reference microRNAs for geometric normalization. Reference microRNAs were selected for their stable expression and confident detection in all pre/post-transplant samples in the discovery cohort as described in Supplement 1. However, these four reference microRNAs were not consistently detected in all samples of the validation cohorts precluding their use for geometric normalization and restricting statistical analysis to RQ instead of NRQ values (Table 1).

Samples from two independent validation cohorts were subjected to blinded central analysis. Validation cohorts additionally contained eight healthy control blood samples to assess specificity. Only 8 of 15 (53%) candidate biomarkers showed clear post-transplant increases in both validation cohorts (RQ values in Table 1): significant (*p* < 0.05) in case of miR-375, miR-132-3p, miR-204-5p, miR-410-3p, miR-200a-3p, miR-429 and miR-125b-5p and a trend for miR-216b-5p.

### 3.5. Verification of Pancreatic Cell Type and Tissue Tropism of the Eight Micrornas Consistently Increased after Islet Transplantation

The pancreatic cell type tropism of these eight microRNAs confirmed in both validation cohorts was additionally analyzed by measuring their relative expression in pancreatic cell fractions enriched in CK19-positive duct cells, amylase-expressing acinar cells and islet endocrine-enriched fractions with lower or higher beta cell number (insulin positivity) and by correlating their post-transplant level to cellular composition of the graft. Whole-body tissue tropism of these microRNAs was additionally analyzed using public data repositories: a comparative analysis of microRNA expression in human islets versus other tissues by next-generation sequencing [10] and a human microRNA tissue atlas, comparing microRNA expression in 61 human tissues to obtain a tissue selectivity index (Appendix A) [22]; the latter data set did not contain islets of Langerhans but did contain pituitary endocrine cells with known similarity to beta cells in terms of selective gene expression [25].

A first set of four microRNAs including miR-375, miR-132-3p, miR-204-5p and miR-410 showed an islet-enriched expression at the whole-body level (Figure 3a upper panel) and a relative beta cell-selective expression within the pancreas (Figure 3a, lower panel). All but miR-410 also showed a statistical correlation to C-peptide content or alpha cell number in the graft (Table 2).

A second set of four microRNAs including miR-200a-3p, miR-429, miR-125b-5p and miR-216b-5p (Figure 3b, lower panel) showed a more predominant expression in pancreatic exocrine cell fractions. Only miR-200a-3p and miR-429 showed relative enriched expression in islets (Figure 3b, upper panel) and correlated to graft C-peptide content (Table 2). miR-216b-5p showed a highly restricted expression in the exocrine pancreas (Figure 3b), clearly indicating acinar cell expression.

In order to be clinically useful as a biomarker of acute beta cell loss in conditions with a higher probability of such a rare event, consistently low to undetectable baseline levels in conditions with a low probability of beta cell loss, are a prerequisite. As shown in Figure 4, miR-375 perfectly meets this criterion, with low to undetectable NRQ/RQ values in all pre-transplant samples (*n* = 23) and all healthy control samples (*n* = 8) of the three study cohorts. The three other islet endocrine-enriched microRNAs, however, show variably detectable levels in pre-transplant samples (in 78% of samples for miR-132-3p, 30% for miR-410) or in healthy control samples (75% of samples for miR-132-3p, 38% for miR-204-5p). The set of four microRNAs with a more predominant pancreatic exocrine expression showed generally low baseline levels both in pre-transplant and healthy control samples, with the exception of miR-125b (Figure 5).

### 3.6. Clinical Validation: Correlation of early Post-Transplant Microrna Levels to Late-Stage Graft Function

Finally, a clinical verification of these eight microRNAs was performed in a third previously reported cohort of 46 T1D patients undergoing islet transplantation [1,9]. In these graft recipients, an elevated GAD65 level (>12.2 pmol/L) and/or miR-375 level (>7.6 pmol/L) 1h after graft infusion, predicted a poor secretory function in these patients 2 months later, defined as a C-peptide increment below 0.5 ng/mL as compared to the fasting C-peptide levels before transplantation [9,12]. Generally, post-transplant GAD65 and miR-375 levels linearly correlate to the C-peptide content of the graft as a proxy of the number of infused beta cells [1,9]. Outliers on these linear regressions—with disproportionately elevated GAD65 or miR-375, as expected from the amount of infused beta cells—generally indicate excessive early graft destruction and poor late-stage outcomes [1]. Figure 6a shows the molar amounts of miR-375 1 h post-transplant, plotted as a function of graft function at 2 months. Patients with outliers for GAD65 (blue dots), miR-375 (green dots) or both (red dots) are indicated in color. Molar amounts of miR-375 correlate well with its RQ values (Figure 6b), thus allowing a similar outlier analysis of the eight candidate biomarker microRNAs at RQ level. As shown in Figure 6c–j, 6 of 7 transplant events with outliers for GAD65 and/or miR-375 also showed outliers for one or more of the other microRNAs. However, none of the other microRNAs was individually superior to miR-375, and their inclusion did not result in the identification of additional poor outcomes, using a C-peptide increment < 0.5 ng/mL as a dichotomous cut-off. Linear regression analysis indicated that besides miR-375, only miR-410 and miR-132 were correlated to graft outcome (Table 2). In multivariate analysis, only miR-132 and miR-375 were retained. ROC analysis confirmed that only miR-375 (AUC = 0.783, 95% CI 0.637–0.891) and miR-132 (AUC = 0.703, 95% CI 0.550–0.828) had significant diagnostic power to predict poor graft outcome at 2 months (Figure 7a). The various microRNAs showed a high degree of redundancy as shown by the correlogram in Figure 7b. In multiple regression, only miR-375 (*p* = 0.0028) was retained as an independent predictor of poor outcome.

## 4. Discussion

Here, we conducted a comprehensive profiling of the microRNA landscape in plasma in a model of acute synchronous necrotic islet cell destruction. A remarkably high number of microRNAs (>200) could be detected in pre-transplant control samples. Islet infusion was, however, associated with prominent surges of a highly restricted set of microRNAs. These expectedly included miR-375, but also several other microRNAs with a previously described beta cell-selective expression and/or function. Four microRNAs (miR-132, miR-204, miR-410 and miR-375) showed a clearly beta cell-enriched expression within the cell populations in our graft preparations and a relevant neuroendocrine/islet-enriched expression at whole-body level [10,22,26] as measured by PCR and sequencing. All were previously shown to be functionally important for beta cells. Reference marker miR-375 was the first microRNA reported to specifically regulate glucose-stimulated insulin secretion and beta cell mass [8,27,28]. It is selectively expressed by pituitary cells and alpha and beta cells with only a 2-fold higher expression in beta versus alpha cells. Its biomarker potential additionally derives from its extremely high molar expression levels in the beta cell, with sequencing read counts 10 times higher than the second most abundant microRNAs, miR-7 and let-7a [10,26] corresponding to an estimated 600,000 miR-375 molecules per beta cell as measured by a calibrated assay [9]. miR-132-3p was recently implied as a positive regulator of alpha cell mass and resistance to stress-induced apoptosis in both alpha [29] and beta [30] cell apoptosis in mouse models and also showed a moderate 3-fold enrichment in beta versus alpha cells [31]. miR-410 is upregulated in glucose-responsive murine insulinoma cell clones [32] and belongs to the top-beta cell enriched microRNAs with 17-fold higher levels in beta versus alpha cells [31]. However, the most exciting microRNA resulting from our selection is miR-204. miR-204 was identified as an important regulator of beta cell function and resistance to endoplasmic reticulum stress: its upregulation by TXNIP-STAT3 signaling leads to suppression of insulin gene transcription via downregulation of MAFA transcription factor [33], and it regulates PERK and ATF4 signaling in beta cells [34] and mediates protection to cytokine-induced ER stress in human beta cells [35]. From a biomarker perspective, miR-204 was the most beta-cell-enriched microRNA with 108-times higher level in human beta versus alpha cells and a relatively abundant read count, albeit 177 times lower than miR-375 [10,26]. Our findings confirm the recent findings by Xu et al. [36] that miR-204 is increased after islet transplantation. More importantly, the latter study also observed increases in plasma miR-204 in the recent-onset phase of T1D, indicating restricted beta cell provenance and highlighting the potential of this microRNA to screen for autoimmune-mediated beta cell destruction.

A second set of four microRNA (miR-200a, miR-429, miR-125b and miR-216b) showed in our analysis a higher relative expression in pancreatic acinar-enriched fractions. One of those, miR-216b, is undoubtedly pancreatic acinar cell-selective: at whole-body level, it is highly enriched in total pancreas tissue, and in rodent models, it was recently reported to be an excellent marker for pancreatitis, superior even to amylase and lipase [37]. For the other three, our observed enrichment in exocrine pancreatic cell fractions conflicts with several reports suggesting endocrine-selective expression and function. miR-200a and miR-429 belong to the co-regulated miR-200a/b/c/142/429 cluster with endocrine-selective expression at whole-body level [38], a relative 2-fold enrichment in beta versus alpha cells [31]. This cluster is upregulated in glucose-responsive murine insulinoma clones [32] and protects beta cells from stress-induced apoptosis in mouse models [39]. Finally, miR-125b was identified as one of the most beta cell-enriched microRNAs, superior even to miR-410 and miR-132. It is possible that miR-125b and miR-200a/429 are indeed endocrine-expressed but that their expression is upregulated ex vivo, in the small subset (1.5%) of beta cells that is co-purified with exocrine cell fractions, resulting in cell stress by the isolation or culture procedures.

In a third independent cohort of 46 transplant events, we then evaluated the diagnostic performance of these eight microRNAs to predict late-stage secretory graft outcome 2 months later. These findings were disappointing: though we observed a clear co-discharge of several newly identified microRNAs with miR-375, only miR-132 showed significant prognostic power (95% confidence interval around AUC excluding AUC = 0.5) but in regression analysis, only miR-375 was retained. This indicates that the newly discovered microRNAs have no added value over miR-375 to predict late-stage graft outcome and are likely inferior to miR-375 as a biomarker of beta cell loss. However, this does not completely erode their diagnostic potential: the addition of the sixendocrine-enriched microRNA (miR-132/204/410/200a/429/125b) to miR-375 in a multiplex- or panel-based PCR in future studies could still be useful to increase the diagnostic specificity in studies screening for possible triggers of nutrients- or cytokine-induced beta cell death.

The strengths of our study are its rigorous design with (i) a stepwise selection of putatively beta cell-enriched microRNAs in well-characterized islet transplant models, allowing the study of correlation between biomarker profiles and cellular graft composition, size and outcome; (ii) the use of several independent validation cohorts originating from two different clinical centers, with centralized but blinded analysis, thus increasing the robustness of our biomarker prioritization; and (iii) the use of a sufficiently sized cohort for clinical validation of their prognostic potential of graft outcome. Cross-comparison with recent literature indicates that this crystallized into a highly relevant panel with siven islet endocrine-selective microRNAs and one exocrine-selective microRNA that requires further study for its potential for early diagnosis of pancreatitis.

Our study also had limitations. Technically we failed to identify proper housekeeping microRNAs for data normalization in the targeted RT-PCRs. Despite their vast biomarker potential, microRNAs have not yet made it into the routine clinical diagnostic lab and independent reproduction of study findings remains challenging. Bench-to-bed translation of microRNAs will require the development of calibrated assays for absolute quantification of circulating microRNA levels or absolute quantification via digital PCR methods. A second limitation for biomarker discovery might be the choice of the model: acute early islet graft destruction likely involves necrotic cell death of endocrine, acinar and ductal cells alike and thus only serves as a rough proxy for the pure beta cell death in autoimmune T1D. Furthermore, it is not clear if gene expression patterns in isolated, cultured beta cells are a reliable approximation of the phenotype of pro-apoptotic, stressed beta cells during the preclinical phase of T1D. It is thus possible that stressed beta cells, around the clinical onset of T1D, upregulate microRNAs involved in endoplasmic reticulum stress responses such as miR-204, and this might boost their biomarker potential by increasing the clinical sensitivity. This could explain why miR-204 could be detected around the onset of T1D [36] but was clearly inferior to miR-375 in our transplant cohorts.

More research is needed to evaluate if combinatorial profiling of miR-375 with miR-132/204/410/200a/429/125b has potential for sensitive detection of occult beta cell death in the preclinical phase and around the onset of T1D. In our opinion, microRNAs have significant potential as compared to both protein-type or (un)methylated DNA markers. Though relatively low throughput to execute, PCR assay development is easy as compared to the selection of high affinity-antibody couples for sandwich immunoassays. Their half-life in plasma (ranging from 1.5 to 13 h [40]) is also relatively favorable with extended circulation likely due to microRNA binding to the 97kDa Argonaute-2 protein or association to extracellular vesicles. As compared to beta cell-selective patterns of (un)methylated DNA, microRNAs have the benefit of their vastly higher molar abundance per cell, and also their longer half-life, with short (170 bp) circulating cell-free DNA fragments being typically cleared within 15 min after discharge.

In conclusion, our study shows that at least six endocrine-enriched microRNAs are co-released with miR-375 from damaged islet grafts. Individually, these newly identified microRNAs are inferior to miR-375 to detect acute synchronous beta cell destruction, quantify early graft destruction and predict late-stage graft function.

## Figures and Tables

**Figure 1 cells-10-01693-f001:**
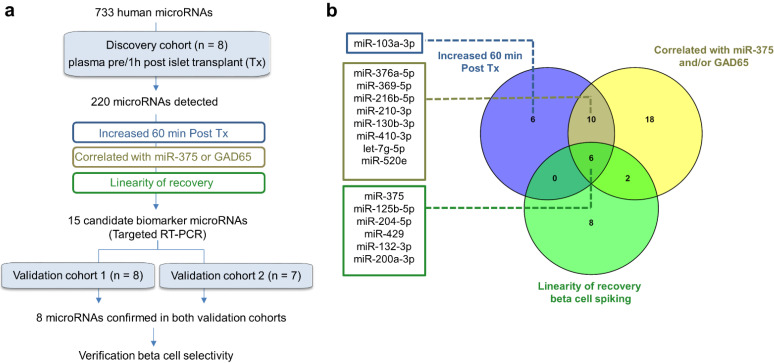
Study design. (**a**). Plasma samples before and 1 h after intraportal islet transplantation (Tx) were arrayed by hydrolysis probe PCR for presence and relative level of 733 human microRNAs in a discovery cohort of 8 subjects transplanted in Brussels. After stringent data filtering, 220 microRNAs were confidently detected and analyzed for differential levels before and after transplantation. A further selection of likely beta cell-enriched microRNAs was done by identifying microRNAs that showed NRQ levels that correlated with plasma GAD65 and/or miR-375 as measured by calibrated quantitative assays; and/or that show a linear response to the number of beta cells spiked into a control plasma pooled sample in a recovery experiment. This led to the composition of a panel of 15 candidate biomarker microRNAs that was subjected to blinded verification by targeted hydrolysis-probe PCR in two independent validation cohorts of 8 patients transplanted in Brussels (validation cohort 1) and 7 patients transplanted in Milan (validation cohort 2). (**b**). The Venn diagram shows the 22 microRNAs that were significantly increased after islet transplantation in the discovery cohort, the 16 microRNAs that increased proportionately to the number of beta cells spiked in the recovery experiment and the 36 microRNAs of which post-transplant NRQ correlated positively with miR-375 and/or GAD65. The 15 microRNAs selected as candidate biomarkers for blinded validation are listed left of the Venn diagram. Finally, microRNAs confirmed in the validation cohorts were additionally analyzed for possible beta cell selectivity by tissue-comparative PCR analysis.

**Figure 2 cells-10-01693-f002:**
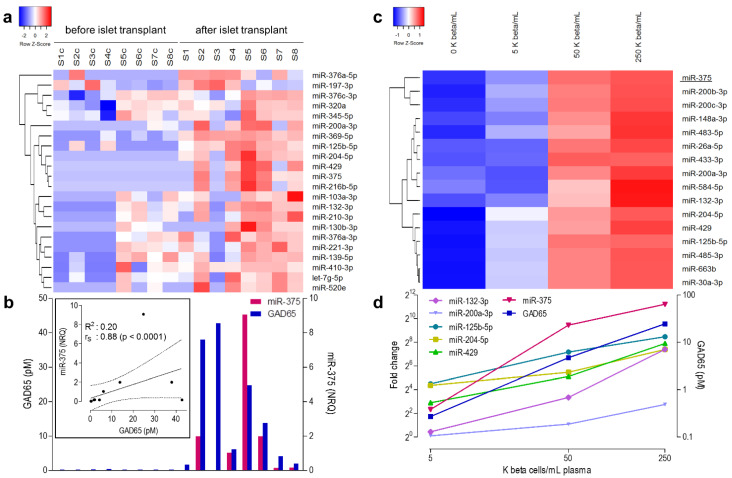
microRNA profiling after islet cell transplantation and beta cell recovery experiment. A total of 733 human microRNAs were measured before and 1 h after intraportal islet cell transplantation in a discovery cohort of 8 transplant events. (**a**). Twnety-two microRNAs showed at least 4-fold increase after islet cell transplantation. Log(NRQ) values were hierarchically clustered and for every microRNA the row z-score is shown. Sc/S indicates the paired samples before and after islet infusion. (**b**)**.** Bar graphs indicate the GAD65 (pM) and miR-375 (NRQ) levels before and one hour after islet cell transplantation and their linear correlation (Inset, *r_S_* = 0.88; *p* < 0.0001). The full line represents a regression line and the dashed lines their 95% confidence interval. (**c**). Sixteen microRNAs increased linearly after spiking of ascending amounts of human beta cell lysate to a healthy control plasma pool (0 K beta cells/mL; 5 K beta cells/mL; 50 K beta cells/mL and 250 K beta cells/mL, 1 K = 10 [3] beta cells). Log (NRQ) values were hierarchically clustered (color key according to z score). (**d**). Fold change with control levels (2^ΔCt^) are shown versus the number of spiked beta cells (log-scale).

**Figure 3 cells-10-01693-f003:**
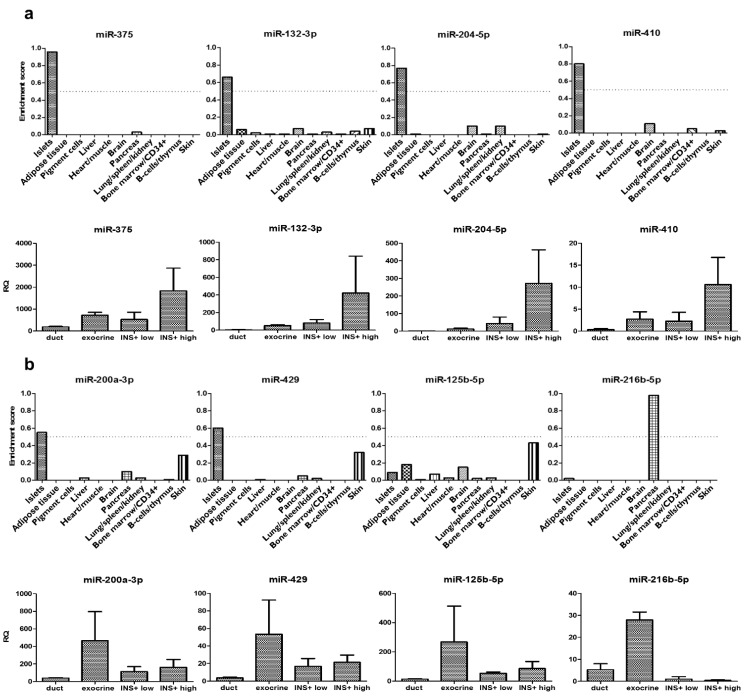
Relative expression levels in human pancreatic cell fractions. (**a**). set of 4 microRNA with a relative islet endocrine-enriched expression and (**b**) set of 4 microRNA with a relative pancreatic exocrine cell-enriched expression. For each set, the upper panel indicates a tissue-comparative enrichment score obtained by next-generation sequencing in human pancreatic islets (*n* = 3 isolates from 6 organs) versus the other indicated tissues as derived from a public data set [10] and the lower panel the RQ measured in this study on cell preparations of human pancreas (all from *n* = 3 independent organ preparations), enriched in duct cells, exocrine acinar cells and islet endocrine-enriched fractions with lower (INS + low) or higher (INS + high) percentage insulin-positive cells. These fractions respectively contained 0.4%, 1.5%, 8.3% and 61.0% insulin-positive cells.

**Figure 4 cells-10-01693-f004:**
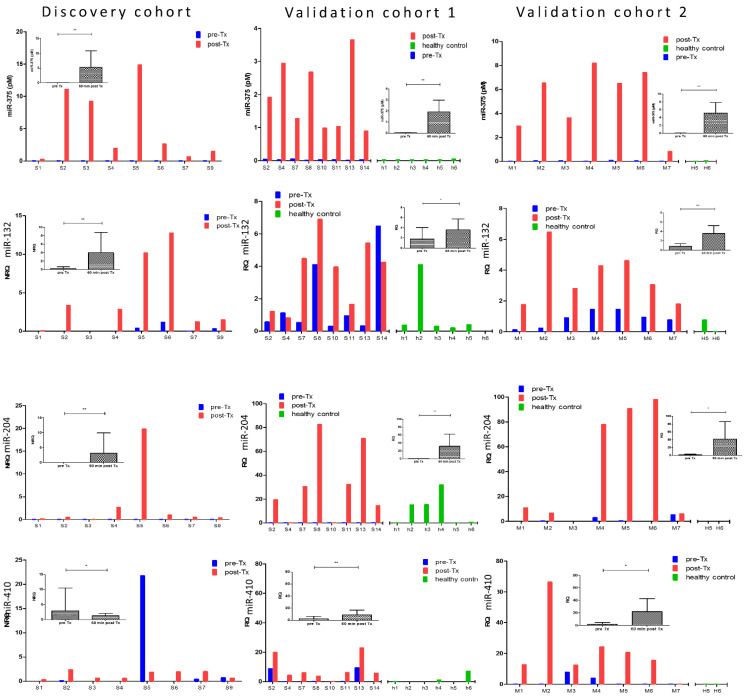
microRNA levels before and 1 h after islet transplantation in three independent cohorts of 4 microRNAs with the strongest islet endocrine enrichment. From top to bottom, panels show the NRQ microRNA levels in individual samples the discovery cohort (*n* = 8 transplants, Brussels) and RQ microRNA levels in validation cohort 1 (*n* = 8 transplants and *n* = 6 healthy controls h1-h6, Brussels) and validation cohort 2 (*n* = 7 transplants and *n* = 2 healthy controls, H5-H6, Milan). Average ± SD and *p* value (* *p* < 0.05; ** *p* < 0.001, paired student-*t* test) in inset. Bar colors indicate pre-transplant (pre-Tx, blue), 1 h post-transplant (post-Tx, red) or healthy control (green).

**Figure 5 cells-10-01693-f005:**
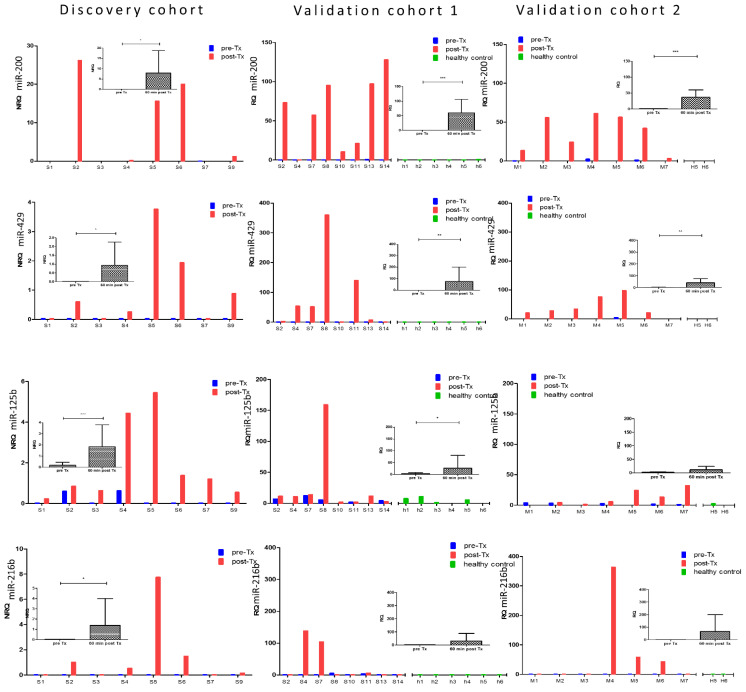
microRNA levels before and 1 h after islet transplantation in three independent cohorts of 4 microRNAs with lower islet endocrine enrichment. From top to bottom, panels show the NRQ microRNA levels in individual samples in the discovery cohort (*n* = 8 transplants, Brussels) and RQ microRNA levels in the validation cohort 1 (*n* = 8 transplants and *n* = 6 healthy controls h1-h6, Brussels) and validation cohort 2 (*n* = 7 transplants and *n* = 2 healthy controls, H5-H6, Milan). Average ± SD and *p*-value (* *p* < 0.05; ** *p* < 0.005; *** *p* < 0.001, paired student *t*-test) in inset. Bar colors indicate pre-transplant (pre-Tx, blue), 1 h post-transplant (post-Tx, red) or healthy control (green).

**Figure 6 cells-10-01693-f006:**
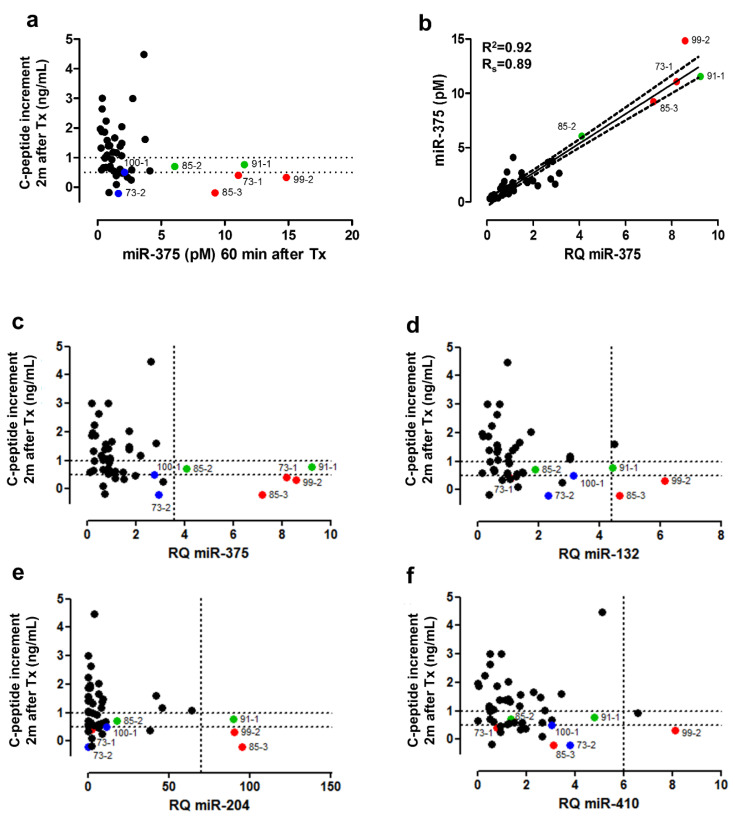
Diagnostic performance of microRNAs to predict poor secretory graft outcome 2 months post-transplantation. (**a**). Molar miR-375 levels (pmol/L) in plasma as measured by calibrated RT-PCR assay 1 h after islet graft infusion in *n* = 46 T1D recipients (*X*-axis) as a function of the graft-induced gain in endogenous C-peptide production (C-peptide increment). Dotted lines indicate operator-chosen thresholds for acceptable graft function at C-peptide increment of 0.5 ng/mL and 1 ng/mL. Further analyses were done taking C-peptide < 0.5 ng/mL as a dichotomous indicator of poor outcome. Generally, 1 h post-transplant levels of miR-375 and GAD65 (not shown, [9]) correlate linearly with the number of implanted beta cells and C-peptide content of the graft. Individual post-transplant patient samples that are outliers on these correlations, indicating excessive beta cell destruction, are numbered and marked as colored dots throughout all panels: outliers for GAD65 (blue dots), miR-375 (green dots) or both (red dots). (**b**). Linear correlation (95% confidence interval as dotted lines) of post-transplant miR-375 values expressed as RQ values (see methods) and molar levels measured by calibrated PCR assay. Panels (**c**–**j**): RQ values of the indicated microRNAs 1 h post-transplant for all *n* = 46 transplant events as a function of C-peptide increment 2 months post-transplant. In case the distribution of RQ values of any microRNA shows statistical outliers, then this threshold is marked with a dotted line on the *X*-axis.

**Figure 7 cells-10-01693-f007:**
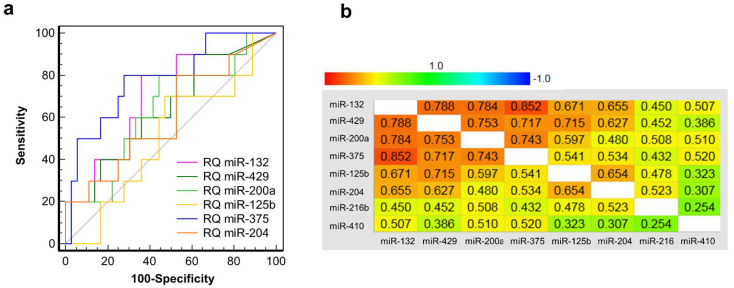
ROC analysis of endocrine-enriched microRNAs 1 h after transplantation to predict secretory graft function 2 months later. (**a**). Area under the receiver operating characteristics (AUC) of the indicated microRNAs in *n* = 46 intraportal islet transplantations, measured 1 h post-transplant, to predict a poor secretory graft function 23 months later (<0.5 ng/mL increment of C-peptide as compared to fasting C-peptide before transplantation). (**b**). Spearman rank correlation coefficients of the indicated microRNAs with a color grading ranging from low (blue) to high (red) correlation.

**Table 1 cells-10-01693-t001:** Panel of 15 candidate microRNA biomarkers of beta cell destruction identified in the discovery cohort and subjected to independent blinded validation. The table lists, from left to right, the microRNA name and corresponding TaqMan^®^ hydrolysis probe used for targeted confirmation, the average Normalized Relative Quantity (NRQ) after global mean normalization, and the Spearman Rank correlation coefficient to plasma GAD65 and miR-375 of the 15 candidate biomarker microRNAs selected, as shown in Figure 1 in the discovery cohort (*n* = 8). Right columns show the average Relative Quantitity (RQ) after versus before transplantation and the corresponding *p*-value (paired student *t*-test) in the independent validation cohort 1 (Brussels, *n* = 8) and validation cohort 2 (Milan, *n* = 7). ND: not detected. Tx: islet transplantation.

		Discovery Cohort (*n* = 8)						Validation Cohort 1 (*n* = 8)		Validation Cohort 2 (*n* = 7)	
		miR Level Post/Pre-Tx		Correlation GAD65		Correlation miR-375		miR Level Post/Pre-Tx		miR Level Post/Pre-Tx	
MicroRNA	Assay ID	NRQ	P	rs	P	rs	P	RQ	P	RQ	P
miR-375	hsa-miR-375-000564	118.1	0.007	0.88	<0.0001	-	-	249.5	0.000	178.0	0.002
miR-132-3p	hsa-miR-132-000457	210.9	0.005	0.50	0.051	0.87	<0.0001	2.0	0.047	4.2	0.004
miR-204-5p	hsa-miR-204-000508	172.0	0.002	0.71	0.002	0.93	<0.0001	315.4	0.003	33.0	0.016
miR-410-3p	hsa-miR-410-001274	49.1	0.034	0.46	0.072	0.57	0.022	3.7	0.002	12.6	0.014
miR-200a-3p	hsa-miR-200a-000502	6249.4	0.044	0.49	0.054	0.82	<0.0001	397.0	0.000	70.4	0.000
miR-429	hsa-miR-429-001024	36.5	0.016	0.64	0.008	0.92	<0.0001	287.0	0.007	53.5	0.002
miR-125b-5p	hsa-miR-125b-000449	52.1	0.001	0.83	0.000	0.79	0.000	6.8	0.039	6.5	0.282
miR-216b-5p	hsa-miR-216b-002326	85.9	0.019	0.66	0.005	0.94	<0.0001	21.6	0.275	191.4	0.084
let-7g-5p	hsa-let-7g-002282	90.5	0.036	0.29	0.274	0.69	0.003	0.6	0.109	1.3	0.650
miR-103a-3p	hsa-miR-103-000439	130.3	0.020	0.18	0.494	0.49	0.054	1.1	0.497	1.2	0.274
miR-130b-3p	hsa-miR-130b-000456	97.1	0.021	0.26	0.336	0.62	0.010	0.8	0.337	1.5	0.086
miR-210-3p	hsa-miR-210-000512	156.7	0.010	0.35	0.188	0.76	0.001	0.0	0.766	0.2	0.317
miR-376a-5p	hsa-miR-376a-5p-002127	9707.1	0.014	0.52	0.040	0.26	0.322	0.0	0.351	2.7	0.115
miR-369-5p	hsa-miR-369-5p-001021	1705.0	0.000	0.93	<0.0001	0.78	0.000	ND		ND	
miR-520e	hsa-miR-520e-001119	167.6	0.027	0.45	0.079	0.75	0.001	ND		ND	

ND: not detected; NRQ: normalized relative quantity; RQ; relative quantiy; Tx: islet transplantation.

**Table 2 cells-10-01693-t002:** Correlation between post-transplant microRNA levels and cellular composition of the islet graft and with secretory graft function 2 months after transplantation. Correlations were analyzed in an independent validation cohort 3 of patients (*n* = 46) receiving at least 2 million beta cells/kg BW, on RQ levels for the indicated microRNAs calculated on paired 1 h post-transplant versus pre-transplant samples for each individual patient. The left panel lists the statistical significance (*p*-value) of Spearman’s rank correlation of RQ to the composition of the grafts in terms of number of insulin-positive cells, C-peptide content, number of glucagon-positive cells, exocrine cells and dead cells measured just prior to implantation. The right panel indicates the correlation of RQ values to the magnitude of the C-peptide increment two months after transplantation [1,9,12] as an indicator of late-stage insulin secretory function of the graft. For correlation with C-peptide and functional graft outcome, multivariate analysis with multiple linear regression was additionally done for parameters retained as significant in univariable analysis.

	Correlation with Graft Characteristics						Functional Outcome	
	Beta Cells	C-Peptide Content		Alpha Cells	Exocrine Cells	Dead Cells	C-Peptide Increment after 2 Months (ng/mL)	
	Spearman (P)	Spearman (P)	Linear Regression (P)	Spearman (P)	Spearman (P)	Spearman (P)	Spearman (P)	Linear Regression (P)
miR-375	0.0541	*p* < 0.0001	0.0001	0.0441	0.0023	0.0424	0.0059	0.0535
miR-132-3p	0.1418	0.0198	0.0037	0.0782	0.0271	0.0262	0.0102	0.0432
miR-204-5p	0.2528	0.0230	0.0074	0.0384	0.5109	0.4291	0.1371	/
miR-410	0.3239	0.1175	/	0.9689	0.0491	0.6401	0.0432	/
miR-200a-3p	0.2320	0.0245	0.0040	0.0859	0.0221	0.0839	0.0828	/
miR-429	0.6479	0.0158	0.0006	0.0778	0.0603	0.6225	0.1249	/
miR-125b-5p	0.5322	0.0328	/	0.0043	0.0900	0.8033	0.1644	/
miR-216b-5p	0.7800	0.1152	/	0.6951	0.3092	0.4530	0.9714	/

## Data Availability

All source data presented in this study are available in Appendix A.

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
