# Peer review of "The MicroRNA Landscape of Acute Beta Cell Destruction in Type 1 Diabetic Recipients of Intraportal Islet Grafts"

_cells, 2021, doi:10.3390/cells10071693_

Round 1

Reviewer 1 Report

The authors have conducted a thoughtfully designed, robust study to understand the detection and reproducibility characteristics of exocrine/endocrine miRNAs in the context of islet transplantation.   The group should be commended for seeking multiple cohorts in which to test their analytes and for investigating multiple biological and technical factors that may impact miRNA detection and utility.  In particular, the work to ensure that qPCR experiments were conducted with appropriate normalization was critical and very much appreciated – while housekeeping miRNAs were not successfully identified, you showed that was the case and found another valid analytical option. I have very few comments on this useful and careful analysis.

  • The authors note that analysis of miRNA in the independent validation cohorts was conducted in a blinded fashion. Further description of how blinding was done/who was blinded would be helpful.
  • Regarding downselection of miRNAs from the 220 that were confidently detected to the 15 followed up: at lines 294-300, a somewhat complicated algorithm for selection is described. However, the Venn diagram indicates that every miRNA that was increased at 60 min post transplant was selected, regardless of the other characteristics.  You may consider simplifying this description. 
  • Table 2’s correlations with graft characteristics was helpful. Did you correlate the miRNAs with the estimated total number of islets transplanted? It seems that some of the miRNA (miR375 for example) correlate with all of the cell types and with dead cells indicating these measures all may be surrogates for the total size of the transplant.
  • At line 404-7, the authors indicate that miRNAs other than miR375 were able to identify 2 additional poor outcome transplants that were not identified by miR375 or GAD65. How many total transplants were considered to have poor outcomes (that is, 2 additional individuals may be a reasonable additional percentage in the population of those transplanted). This would also help understand the data from miR375 and GAD65 better.
    • A minor note on Figure 6 – the blue dots were nearly invisible when printed. You might consider changing the color slightly.

Author Response

Reviewer 1 - Cells-1228965

Comments and suggestions for authors

The authors have conducted a thoughtfully designed, robust study to understand the detection and reproducibility characteristics of exocrine/endocrine miRNAs in the context of islet transplantation.   The group should be commended for seeking multiple cohorts in which to test their analytes and for investigating multiple biological and technical factors that may impact miRNA detection and utility.  In particular, the work to ensure that qPCR experiments were conducted with appropriate normalization was critical and very much appreciated – while housekeeping miRNAs were not successfully identified, you showed that was the case and found another valid analytical option. I have very few comments on this useful and careful analysis.

Authors’ reply: we thank Reviewer 1 for this constructive comment and overall appreciation. Evidently, when starting our study we hoped to identify microRNAs that were superior or at least complementary to the established marker miR-375. Even though the conclusion of our descriptive study shows that this is not the case, we feel that is useful to share our data with other researchers pursuing a similar aim of finding optimal diagnostic solutions for sensitive detection of pancreatic beta cell injury.

  • The authors note that analysis of miRNA in the independent validation cohorts was conducted in a blinded fashion. Further description of how blinding was done/who was blinded would be helpful.

Authors’ reply: For blinding, all pre-and post-transplant patient samples and healthy controls received a random numerical code within each cohort. All samples were then centrally analyzed by PCR and only after finalization of data processing with normalization, samples were unblinded. We have now stated this more explicitly in Methods line 103-106.

  • Regarding downselection of miRNAs from the 220 that were confidently detected to the 15 followed up: at lines 294-300, a somewhat complicated algorithm for selection is described. However, the Venn diagram indicates that every miRNA that was increased at 60 min post-transplant was selected, regardless of the other characteristics.  You may consider simplifying this description. 

Authors’ reply: the step-wise down-selection is described on lines 216-232. We agree that our selection process is rather complex and though we outlined the various strategies of selection, the final down-selection from 22 upregulated microRNAs post-transplant to 15 candidate biomerker microRNAs selected for further validation is in part operator dependent. We admit that we might have lost valuable candidate microRNA markers in the down-sizing from 22 to 15. The further a posteriori analysis of beta cell selectivity that was described on lines 294-300 had no influence on the selection. We have rephrased this sub-selection process on lines 225-234 and clearly state that we made a manual selection, guided by the 3 mentioned criteria. We have also adapted Fig. 1 and more clearly stated that 8 microRNAs were subsequently confirmed in both validation cohorts, and then that these were subjected to a posteriori analysis of beta cell selectivity. We feel that these changes more clearly describe the sub-selection process.

  • Table 2’s correlations with graft characteristics was helpful. Did you correlate the miRNAs with the estimated total number of islets transplanted? It seems that some of the miRNA (miR375 for example) correlate with all of the cell types and with dead cells indicating these measures all may be surrogates for the total size of the transplant.

Authors’ reply: the islet transplant protocol at University Hospital Brussels (Center for Beta Cell Therapy in Diabetes) does not quantify islet equivalents (IE) using dithizone – as in done in many centers – but performs a standard combination of immunofluorescence microscopy (insulin, glucagon), electrom microscopy (granulation state) in combination with a propidium iodide vital staining for % living cells. Reviewer 1 is correct that microRNA-375 indeed correlates with total number of endocrine cells in the graft. This was clearly presented in our prior studies (Ling et al., 2015; Roels et al., 2019) on GAD65 and miR-375: in both studies we showed that the post-transplant surge was linearly correlated to the total graft size, and correlated even more closely to total number of insulin-positive cells. Individual transplants that evolved towards poor outcome were characterized by post-transplant miR-375/GAD65 that were disproportionately elevated as compared to the number of implanted insulin+ cells (outliers on this linear regression, as shown in Fig. 6 in this manuscript). The latter thus indicates that use of those microRNAs should either always be after normalization for each individual patient for the number of implanted cells. Since this is clinically not practical, we proposed an leaner approach in our latest manuscript (Roels et al., 2019) by introducing a dual threshold with a gray zone. We did not include such analysis in this descriptive/discovery study to avoid making it even more complex to read, and also because none of the newly identified microRNAs showed complementary diagnostic power as compared to microRNA-375.

We also addressed the comment by Reviewer 1 statistically, by analyzing if normalization of post-transplant microRNA levels (RQ) by dividing them by the total cell number of the graft could improve diagnostic performance for outcome prediction (see statistical analysis below), but this was not the case.

  • At line 404-7, the authors indicate that miRNAs other than miR375 were able to identify 2 additional poor outcome transplants that were not identified by miR375 or GAD65. How many total transplants were considered to have poor outcomes (that is, 2 additional individuals may be a reasonable additional percentage in the population of those transplanted). This would also help understand the data from miR375 and GAD65 better.

Authors’ reply: we revisited the data that were graphically shown in Fig. 6. The sentence to which Reviewer 1 refers was in fact an error. Using C-peptide increment of < 0.5 ng/mL as dichotomous indicator of poor outcome, no additional poor outcome was identified by any other microRNA. In logistic regression only miR-375 RQ independently predicted poor outcome. Also when using the RQ values normalized by the total number of implanted cells, only miR-375 independently predicted poor outcome. We apologize for this mistake, and have corrected our mistake on lines 336-339. Valid point, thank you!

In the uploaded fiele for review we present the multiple logistic regression. As shown below, only RQ miR-375 was retained in multiple regression as independent predictor of poor outcome, defined as C-peptide increment of < 0.5 ng/mL at 2 months after transplantation. We also analyzed if normalization for total cell number in the graft (designated as RQ_miR_XXX_total_cells) could improve diagnostic performance, but the result of this multiple regression analysis was similar, showing that only miR-375 independently predicted poor outcome.

Reviewer 2 Report

The study by Geert A. Martens et al is a descriptive study connecting miRNA levels to acute beta cell destruction in Type 1 diabetic recipients after islet transplantation.

The study is divided in three parts, first they study the population of miRNAs that are regulated in the “discovery cohort” comparing plasma miRNAs levels before and 1-hour after transplantation. Then, candidate miRNAs were validated in two “validation cohorts” in two independent hospitals. They have found that miR-375, which had been previously described as an important miRNA in beta-cell function and mass preservation, and it had been also described as beta cell survival biomarker, was the most important biomarker of beta-cell death. Authors report that the combinatorial profiling of miR-375 with miR-526132/204/410/200a/429/125b has potential for sensitive detection of occult beta cell death in the preclinical phase and around onset of type 1 diabetes. Furthermore, most of the other miRNAs found to be relevant e.g., 200a, 429, 204, … have already been related to beta-cell death.

Although the study is well performed and expected results were promising for the field, at the end, the study is confirmatory. Manuscript major conclusion is that the combination of miR-375 with miR-526132/204/410/200a/429/125b is potentially more sensitive to detect beta-cell death than miR-375 alone, but this point is not demonstrated in the study. Authors should develop this comparative study which would reinforce their major conclusion and it would give novelty to this confirmatory study.

Author Response

Reviewer 2 - Cells-1228965

General: are the conclusions supported by the results -> must be improved

Comments and suggestions for the authors

The study by Geert A. Martens et al is a descriptive study connecting miRNA levels to acute beta cell destruction in Type 1 diabetic recipients after islet transplantation.

The study is divided in three parts, first they study the population of miRNAs that are regulated in the “discovery cohort” comparing plasma miRNAs levels before and 1-hour after transplantation. Then, candidate miRNAs were validated in two “validation cohorts” in two independent hospitals. They have found that miR-375, which had been previously described as an important miRNA in beta-cell function and mass preservation, and it had been also described as beta cell survival biomarker, was the most important biomarker of beta-cell death. Authors report that the combinatorial profiling of miR-375 with miR-526132/204/410/200a/429/125b has potential for sensitive detection of occult beta cell death in the preclinical phase and around onset of type 1 diabetes. Furthermore, most of the other miRNAs found to be relevant e.g., 200a, 429, 204, … have already been related to beta-cell death.

Although the study is well performed and expected results were promising for the field, at the end, the study is confirmatory. Manuscript major conclusion is that the combination of miR-375 with miR-526132/204/410/200a/429/125b is potentially more sensitive to detect beta-cell death than miR-375 alone, but this point is not demonstrated in the study. Authors should develop this comparative study which would reinforce their major conclusion and it would give novelty to this confirmatory study.

Authors’ reply: in part we agree with Reviewer 2’s appreciation of our study as confirmatory. We aimed to discover and validate additional microRNA markers that showed added diagnostic value for beta cell death in vivo, and its clinical use for prognostication of poor late-stage islet graft outcome. Our results were disappointing in the end, since none of the candidate biomarker microRNAs had additional value as compared to microRNA-375. On the other hand, these negative findings strengthen the importance of biomarkers such as microRNA-375 and GAD65 in the model of islet transplantation. We have shown before that both GAD65 and miR-375 are also superior to unmethylated insulin DNA for this purpose (Roels et al., 2019). We found it important to share our step-wise selection and analysis, since also negative data are important to orient the field. Several of the microRNAs that made it to our core set of 8 confirmed candidate biomarkers have been very recently identified by others, as functionally important but also as biomarker of beta cell injury in vivo. This independent confirmation by our study is thus also helpful for the field.

We disagree with Reviewer 2’s appreciation that our conclusions are not supported by the data. In the Discussion section we have very honestly highlighted the strengths but also the limitations of our data. We have clearly stated that none of the additional microRNAs showed additional diagnostic value independent from miR-375. We emphasized that an important limitation of our study design, was the selection of human islet transplantation for selection of candidate biomarkers of beta cell death, in part because the massive synchronous beta cell necrosis likely is fundamentally different from the auto-immune-mediated apoptosis in the preclinical and onset stages of type 1 diabetes. We are currently iterating this analysis in the setting of recent-onset type 1 diabetes in humans and in animal models, and these data will be dealt with in a separate study.

We agree with Reviewer 2 that we could not formally show that addition of miR-526/132/204/410/200a/429/125b could be promising to detect episodes of occult beta cell destruction – the last sentence of the manuscript, Discussion line 441-442 – so we decided to remove this sentence. We hope this is acceptable for Reviewer 2. In our opinion, all the other statements in the Discussion are well supported by our new data and prior work by us and other groups.

Roels, S., Costa, O.R., Tersey, S.A., Stange, G., De Smet, D., Balti, E.V., Gillard, P., Keymeulen, B., Ling, Z., Pipeleers, D.G., et al. (2019). Combined Analysis of GAD65, miR-375, and Unmethylated Insulin DNA Following Islet Transplantation in Patients With T1D. The Journal of clinical endocrinology and metabolism 104, 451-460.

Round 2

Reviewer 2 Report

I cannot give any other feedback because the authors did not provide any new data.